# The characteristics and treatment patterns of patients with Parkinson's disease in the United States and United Kingdom: A retrospective cohort study

Linda Kalilani[1]*, David Friesen[2], Nada Boudiaf[2], Mahnaz Asgharnejad[1]

**1** UCB Pharma, Raleigh, North Carolina, United States of America, **2** UCB Pharma, Slough, United Kingdom

* linda.kalilani@ucb.com

## Abstract

### Objectives

The objective of the study was to describe treatment patterns in patients newly diagnosed with Parkinson's disease (PD) in the United States (US) and the United Kingdom (UK).

### Methods

This retrospective cohort study used the US IBM MarketScan database (2012–2017) and the UK Clinical Practice Research Datalink (CPRD) (2004–2015) database to describe treatment patterns in incident PD cases. Patients fulfilling the case definition of PD, $\geq$30 years, with a 2-year baseline period prior to the index date (date of PD diagnosis), and $\geq$90 days of follow-up were included in the study. Treatment was classified as monotherapy (one PD medication for $\geq$60 continuous days), polytherapy (at least two PD medications concurrently for $\geq$60 days), or untreated (no PD medication prescription). Treatment patterns described included type of medication, duration and outcome of treatment.

### Results

There were 11,280 patients in IBM MarketScan and 7775 patients in CPRD who fulfilled the study criteria. The proportion of treated patients was 62.4% (US) and 78.6% (UK). The majority of patients were prescribed monotherapy as first-line treatment (US: 85.2%, UK: 68.5%). Levodopa was the most frequently prescribed first-line medication (US: 70.1%, UK: 29.0%). There were 57.9% in the US and 23.8% in the UK who remained on the first monotherapy treatment till the end of the study.

### Conclusion

The study has highlighted the current treatment practices in the US and UK, and underscored differences in the two regions impacted by treatment policies and guidelines.

**Data Availability Statement:** Original de-identified data used in this analysis were obtained from and are the property of IBM MarketScan (US data) and

Clinical Practice Research Datalink (UK data). IBM MarketScan and Clinical Practice Research Datalink provided the raw data to UCB Pharma, which were used to create the analytical files for the study. IBM MarketScan and Clinical Practice Research Datalink do not have access to these analytical files and do not have a right to provide them to a third party. UCB Pharma will make the analytical files available to any researcher who requests them from UCB Pharma for non-commercial purposes after obtaining the necessary approval for third party access from IBM MarketScan and Clinical Practice Research Datalink. Any researcher requiring access to the raw data that were used to generate the analytical files can access the data directly through IBM MarketScan and Clinical Practice Research Datalink under a license agreement, including the payment of appropriate license fees, between that third party and IBM MarketScan and Clinical Practice Research Datalink. Researchers interested in accessing the de-identified analytical data files from UCB Pharma for the above stated purposes should contact Linda Kalilani (Linda.Kalilani@ucb.com).

**Funding:** This study was supported by UCB Pharma, Brussels, Belgium. Linda Kalilani is an employee of UCB Pharma. David Friesen (Friesen Limited, Ascot, UK) and Nada Boudiaf (Chiltern International, Slough, UK) were paid consultants for UCB Pharma on this study. Mahnaz Asgharnejad is a former employee of UCB Pharma. The funder provided support in the form of salaries or consultancy fees for authors [LK, MA, DF, NB], who designed the retrospective analysis ('study'), performed the statistical analysis, and decided to publish. The specific roles of these authors are articulated in the 'author contributions' section.

**Competing interests:** Linda Kalilani is an employee of UCB Pharma. David Friesen (Friesen Limited, Ascot, UK) and Nada Boudiaf (Chiltern International, Slough, UK) were paid consultants for UCB Pharma on this study. Mahnaz Asgharnejad is a former employee of UCB Pharma. This does not alter the authors' adherence to the PLOS ONE policies on data sharing and materials.

# Introduction

Parkinson's disease (PD) is a progressive neurological disorder caused by the degeneration of dopaminergic neurons in the substantia nigra, an area of the midbrain that plays a crucial role in voluntary motor control. The incidence rate of PD ranges between 1.5 and 22 per 100,000 person-years [1]. The rate of symptom progression in PD patients is heterogeneous across patients, and throughout the course of disease within the same patient [2]. Approximately 40% to 75% of the patients with PD die after 10 to 20 years of disease history, and 50% of the patients who survive require nursing-home care [3].

Treatment for PD is mainly focused on correcting the dopaminergic deficit by either stimulating dopaminergic receptors to increase the levels of dopamine or by using dopamine agonists, thereby alleviating the cardinal motor symptoms of the disease. Management also includes the treatment of non-motor symptoms, which are common and affect the quality of life of patients with PD [4]. Treatment of PD is individualized with the goal of managing the most severe symptoms with the least amount of medication while promoting functional independence and minimizing short- and long-term adverse effects.

There are numerous treatment options because many PD medications exist currently on the market, and this may complicate treatment decisions. Owing to variability in treatment guidelines and reimbursement policies, comparison of treatment patterns across regions is challenging. Limited data are available from clinical practice that describe the treatment pathways of patients with PD in the United States (US) and United Kingdom (UK) [5,6]. Drug utilization studies provide information on the current treatment trends, and an opportunity to assess whether patients are treated according to recommended treatment guidelines. The objective of this study was to describe the characteristics and treatment patterns of patients with PD in the US and UK.

# Study methods

## Data sources

This retrospective cohort study was conducted using two databases: the US IBM MarketScan database and the UK Clinical Practice Research Database (CPRD) database. The most current data available at time of the study were used.

IBM MarketScan claims database covered the period from January 1, 2012 to September 30, 2017. The database includes over 100 million lives of commercially insured individuals (i.e., working age adults and their dependents), individuals with supplemental Medicare coverage plus employer-paid commercial plans, and data from individuals with limited incomes whose insurance is paid by the state. This database contains administrative claims information on medical treatment (inpatient, outpatient, and emergency care), outpatient pharmacy prescriptions, and enrollment history.

The CPRD was linked to the Hospital Episode Statistics (HES) database, and covered the period from January 2004 to September 2015. The CPRD consists of routinely collected anonymized longitudinal electronic health record data from approximately 690 primary care practices in the UK. The CPRD database contains information on patient demographics, symptoms, clinical diagnoses, laboratory test results, prescriptions of medicine, health-related behaviors, and referrals to secondary care (hospitals or specialists). The HES inpatient database provides information on inpatient care (including patients admitted through the accident and emergency department) delivered by NHS hospitals in England. Data including basic demographics, clinical diagnoses, procedures, and administrative information are captured in HES.

## Study design

Incident PD cases were identified in the IBM MarketScan database between January 1, 2014 and June 30, 2017. Patients were defined as having a PD diagnosis if they fulfilled at least one of the following criteria:

- The presence of at least one inpatient claim with the ICD-9-CM code 332.0, in any field of diagnosis

- At least two outpatients claim with the ICD-9-CM code 332.0 (occurring at least 30 days apart and within 365 days) in any field of diagnosis

- At least one outpatient claim with the ICD-9-CM code 332.0 in any field of diagnosis plus at least two prescription claims for PD-related medication (levodopa-carbidopa, anticholinergics, dopamine agonist, MAO-B inhibitors, or COMT inhibitors) within 6 months following the diagnostic code for PD.

In the CPRD, incident PD cases were identified between 2006 and 2015. Patients were defined as having a PD diagnosis if they fulfilled at least one of the following criteria:

- An HES-recorded inpatient visit with an ICD-10 code indicating a diagnosis of PD (G20.xx)

- A primary care read code indicating a diagnosis of PD, where the diagnosis date was the primary care consultation date at which the read code was recorded

- A primary care read code indicating primary care-recorded evidence of a neurologist consultation with a diagnosis of PD, where the diagnosis date was the consultation date at which the read code is recorded.

Additionally, all study participants were required to be aged at least 30 years, and have a 2-year baseline period with continuous medical enrollment with medical and insurance coverage in the IBM MarketScan database, or to be continuously registered at a practice for at least 2 years prior to the PD diagnosis in the CPRD database. Patients were excluded if they had a diagnosis of PD, secondary PD or parkinsonism (including drug-induced PD, vascular PD, essential tremor, malignant neuroleptic syndrome, postencephalytic parkinsonism, syphilitic parkinsonism), and dementia; any prescription claims for PD-related medication (levodopa-carbidopa, anticholinergics, dopamine agonists, MAO-B inhibitors, or COMT inhibitors) during the baseline period; and any of the following prescriptions: phenothiazines, butyrophenones, flupentixol, metoclopramide, reserpine, amiadarone and cinnarizine within 180 days before the index date, and a follow-up period of less than 90 days.

Patients in the IBM MarketScan database were followed from the index date until the first occurrence of any of the following: date of last enrollment after the index date for participants who were not continuously enrolled through the end of the study period, or end date of the study period for participants continuously enrolled through the end of the entire observation period. For the CPRD database, patients were followed to the first occurrence of any of the following: the end of patient registration including transfer out of the patient from the practice and death, the last data collection from the practice (i.e. end of data coverage), or the end of the study period (CPRD data availability based on the latest update).

With the IBM MarketScan database, the data were previously collected and statistically de-identified and are compliant with the conditions set forth in Sections 164.514(a)-(b)(1)ii of the Health Insurance Portability and Accountability Act of 1996 Privacy Rule; therefore, approval from an institutional review board was not sought.

With the CPRD database, the data were previously collected and statistically de-identified and are compliant with the conditions set forth in Sections 164.514(a)-(b)(1)ii of the Health Insurance Portability and Accountability Act of 1996 Privacy Rule; therefore, approval from an institutional review board was not sought. The protocol was reviewed and approved by an Independent Scientific Advisory Committee, a non-statutory expert advisory body established in 2006 by the Secretary of State for Health to provide scientific advice on research requests to access data provided by CPRD.

**Study outcomes.**   The treatment patterns evaluated included duration of treatment, type of prescription (monotherapy or polytherapy), outcome of the treatment (discontinued, switched, augmentation, or continued on treatment). Patients were defined as having received PD treatment if they had prescriptions with at least 30 days of supply during follow-up. Patients were classified as untreated if they had no prescription record of a PD medication any time after the index date.

Treatment was classified as monotherapy (prescription claims for one PD medication for ≥60 days), polytherapy (simultaneous prescription claims for at least two PD medications for ≥60 continuous days or formulations with more than one PD medication with different mechanisms of action), or undetermined (either PD medication claims for <60 days or claims started within 60 days in the UK and 90 days in the US of the end of the follow-up period). The duration of a PD treatment was defined as the number of days from the first to last PD dispensing record plus the number of days of supply following the date of the last dispensing record, including gaps in treatment less than 60 days in the UK and 90 days in the US. A treatment line was defined as a period during which the patient was treated with a constant PD medication regimen. The first-line treatment commenced upon prescription of the first PD medication on or after the index date. PD treatment regimen outcomes were defined as continued (no treatment change until end of follow-up period); augmented (addition of at least one PD medication to an existing PD treatment regimen); switched (current treatment line contained PD medication not in the next line and next line contained PD medication not present in the current line); or discontinued (no prescription for the PD medication after the last prescription of >60 days). The patient characteristics evaluated included age and gender defined at index date, and comorbidities evaluated in the 1-year period before the index date.

**Statistical analysis.**   Descriptive statistics were used to summarize the characteristics of the study subjects and the treatment patterns. Continuous variables were presented as means and standard deviation (SD) if normally distributed, or median, interquartile range (IQR), and range if not normally distributed. Categorical variables were presented as frequencies and percentages. The analysis was conducted using SAS version 9.4 software (SAS Institute, Inc, Cary, North Carolina).

## Results

### Study participants

Of the 113,647 patients who had at least one diagnosis code for PD in the IBM MarketScan database, 11,280 fulfilled the study criteria and were included in the US study. The mean (standard deviation [SD]) age of the patients was 73.0 (12.0) years, and 59.6% were male. The median (range) duration of follow-up was 465.0 (90–1369) days.

Of the 37,965 patients who had at least one diagnosis code for PD in the CPRD database, 7775 fulfilled the study criteria and were included in the UK study. The mean (SD) age was 73.8 (10.3) years and 62.6% were male. The median (range) duration of follow-up was 1006 (90–3797) days.

Table 1 provides details of characteristics of the patients including the most prevalent comorbidities and frequently prescribed medications in the year prior to the diagnosis of PD.

**Table 1. Demographics and baseline characteristics of the study participants.**

| Patient Characteristics | IBM MarketScan Database, N = 11,280 | Patient Characteristics | CPRD Database, N = 7775 |
|---|---|---|---|
| Age (years), mean (SD) | 73.0 (12.0) | Age (years), mean (SD) | 73.8 (10.3) |
| Male, n (%) | 6718 (59.6) | Male, n (%) | 4855 (62.4) |
| **Prevalence of comorbidities,[a] n (%)** | | **Prevalence of comorbidities,[a] n (%)** | |
| Hypertension | 4378 (38.8) | Primary hypertension | 684 (8.8) |
| Hyperlipidemia | 3377 (29.9) | Constipation | 634 (8.2) |
| Mobility impairment related comorbidities | 2981 (26.4) | Accidental falls | 432 (5.6) |
| Arthritis | 2547 (22.6) | Type 2 diabetes Mellitus | 405 (5.2) |
| Pain in joint | 2432 (21.6) | Urinary tract infection | 383 (4.9) |
| Type 2 diabetes mellitus | 2331 (20.7) | Ischemic heart disease | 360 (4.6) |
| Malaise and fatigue | 2225 (19.7) | Shoulder pain | 335 (4.3) |
| Pain in limb | 1992 (17.7) | Dementia | 282 (3.6) |
| Coronary atherosclerosis | 1913 (17.0) | Low back pain | 282 (3.6) |
| Dyspnea and respiratory abnormalities | 1694 (15.0) | Atrial fibrillation and flutter | 258 (3.3) |
| **Prevalence of medication use,[b] n (%)** | | **Prevalence of medication use,[b] n (%)** | |
| Atorvastatin Calcium | 2388 (21.2) | Aspirin 75 mg dispersible tablets | 1897 (24.4) |
| Acetaminophen/Hydrocodone Bitartrate | 2165 (19.2) | Paracetamol 500 mg tablets | 1658 (21.3) |
| Levothyroxine Sodium | 1956 (17.3) | Simvastatin 40 mg tablets | 1255 (16.1) |
| Lisinopril | 1903 (16.9) | Bendroflumethiazide 2.5 mg tablets | 1068 (13.7) |
| Amlodipine Besylate | 1754 (15.5) | Omeprazole 20 mg gastro-resistant capsules | 1103 (14.2) |
| Ciprofloxacin Hydrochloride | 1559 (13.8) | Lactulose 3.1–3.7 g/5 ml oral solution | 772 (10.0) |
| Omeprazole | 1557 (13.8) | Trimethoprim 200 mg tablets | 771 (9.9) |
| Simvastatin | 1557 (13.8) | Amoxicillin 500 mg capsules | 735 (9.5) |
| Furosemide | 1553 (13.8) | Amlodipine 5 mg tablets | 696 (9.0) |
| Metformin Hydrochloride | 1420 (12.6) | Co-codamol 8 mg/500 mg tablets | 679 (8.7) |
| **Health Care Utilization, n (%)** | | **Health Care Utilization, n (%)** | |
| At least 1 inpatient visit | 2664 (23.6) | At least 1 inpatient visit | 1625 (20.9) |
| At least 1 outpatient visit | 11,136 (98.7) | At least 1 outpatient visit | 5830 (75.0) |
| At least 1 emergency department visit | 4040 (35.8) | At least 1 emergency department visit | 1741 (22.4) |

The prevalence of comorbidities and medications were measured in the 1-year period prior to the index date.

SD, standard deviation.

[a]The top 10 most prevalent comorbidities identified in each database.

[b]The top 10 medications most frequently prescribed medications in each database.

## First-line treatment

In the US, 4241 (37.6%) of the patients with PD did not have documented PD medication prescriptions filled during the study period, whilst 7039 (62.4%) patients received PD treatment during the follow-up period. Of the treated patients, 85.2% and 1.3% were prescribed monotherapy and polytherapy, respectively, for ≥60 days as first-line treatment. The remaining 13.4% of treated patients comprised those receiving treatment for <60 days or having started treatment within 60 days of the end of follow-up and therefore were classified as undefined or indeterminate.

In the UK, 1658 (21.4%) patients with PD did not have documented PD medication prescriptions during the study period, whilst 6097 (78.6%) patients received PD treatment prescriptions during the follow-up period. Of the treated patients, 68.5% and 16.2% were prescribed monotherapy and polytherapy, respectively, for ≥60 days as first-line treatment. The remaining 15.3% of treated patients comprised those receiving treatment for <60 days (undetermined).

**Table 2. The most common first-line PD medication for US patients in the IBM marketscan database[a].**

| Initial PD Medication | N (%) | Median (min-max) days from first- to second-line[b] | Continued[c] | | Augmented[d] | | Switched[e] | | Discontinued[f] | |
|---|---|---|---|---|---|---|---|---|---|---|
| | | | % of patients | Median (min, max) days to event | % of patients | Median (min, max) days to event | % of patients | Median (min, max) days to event | % of patients | Median (min, max) days to event |
| Levodopa | 4932 (70.1) | 250.0 (1–1366) | 59.0 | 342.0 (91–1366) | 12.3 | 115.0 (1–1165) | 2.9 | 100.0 (8–887) | 23.5 | 153.0 (33–1206) |
| Rasagiline | 579 (8.2) | 156.0 (1–1256) | 32.6 | 270.0 (91–1159) | 39.6 | 89.0 (1–1050) | 11.4 | 92.0 (7–539) | 8.8 | 174.0 (52–979) |
| Ropinirole | 357 (5.1) | 164.0 (1–1198) | 32.8 | 257.0 (91–1198) | 27.2 | 133.0 (1–1185) | 15.7 | 116.0 (15–627) | 20.4 | 130.0 (37–665) |
| Pramipexole | 324 (4.6) | 189.0 (2–1250) | 37.7 | 260.5 (92–1250) | 24.7 | 118.5 (2–867) | 14.5 | 113.0 (28–693) | 16.4 | 141.0 (34–990) |
| Amantadine | 169 (2.4) | 120.0 (1–843) | 30.2 | 184.0 (91–779) | 23.1 | 73.0 (1–580) | 20.7 | 107.0 (31–843) | 22.5 | 90.0 (35–332) |
| Rotigotine | 79 (1.1) | 127 (19–923) | 21.5 | 174.0 (94–906) | 40.5 | 109.0 (19–886) | 20.3 | 118.5 (19–489) | 13.9 | 90.0 (58–516) |

N, total number of patients; PD, Parkinson's disease.

[a]The table includes medications that were prescribed in ≥1% of the patients with PD.

[b]Median time is reported for the overall population, including patients progressing to second-line treatment and those who continued on first line treatment through the end of the study period.

[c]Continued indicates no treatment change until end of follow-up for the patient.

[d]Augmented indicates the addition of at least one PD medication to the current PD regimen.

[e]Discontinued indicates no prescription for the PD medication after the last prescription of >60 days.

[f]Remaining percentage that is not included in the table represents undefined and indeterminate cases, representing patients who received treatment<60 days or started treatment within 90 days of end of follow-up, respectively.

## Most common PD medications used for first-line treatment

The most commonly prescribed PD medications and progression to second-line treatments for US patients are listed in Table 2. Levodopa was the most common first-line PD treatment, prescribed to 70.1% of treated patients. Other commonly prescribed PD medications were rasagiline, ropinirole, pramipexole, and amantadine. More than half of patients were pre-scribed levodopa, and just over a third of patients who were prescribed rasagiline, ropinirole, pramipexole and amantadine as initial treatment remained on this first-line monotherapy dur-ing the study. The median time to second-line treatment for these commonly prescribed PD medications ranged from 120 to 250 days.

The most commonly prescribed PD medications and progression to second-line treatments in the UK are listed in Table 3. Levodopa was also the most common first-line PD treatment in the UK, prescribed to 29.0% of treated patients. Other commonly prescribed PD medications were pramipexole, entacapone combined with levodopa, ropinirole, and pergolide. Almost a quarter of the patients prescribed levodopa as initial treatment remained on this first-line monotherapy during the study. The median time to second-line treatment for the most com-monly prescribed first-line PD medications ranged from 60 to 329 days.

## Progression from first-line to next-line treatment (treatment additions or switches)

For the US study population (N = 11,280), 37.6%, 34.0%, 16.4%, 6.6%, 2.7%, and 2.7% of patients received none, one, two, three, four, and at least five treatment lines, respectively, over a median follow-up of 465.0 (range 90–1369) days. Of the 5998 patients receiving first-line

**Table 3. The most common first-line PD medication for UK patients in the CPRD database[a].**

| Initial PD Medication | N (%) | Median (Q1, Q3) days from first- to second-line[b] | Continued[c] | | Augmented[d] | | Switched[e] | | Discontinued[f] | |
|---|---|---|---|---|---|---|---|---|---|---|
| | | | % of patients | Median (Q1, Q3) days to event | % of patients | Median (Q1, Q3) days to event | % of patients | Median (Q1, Q3) days to event | % of patients | Median (Q1, Q3) days to event |
| Levodopa | 1768 (29.0) | 329 (125–739) | 24.4 | 606 (310–1033) | 40.4 | 275 (95–669) | 10.1 | 186 (94–368) | 17.6 | 216 (99–471) |
| Pramipexole | 1235 (20.3) | 271 (97–624) | 20.8 | 515 (287–965) | 20.2 | 274 (97–603) | 29.4 | 134 (60–321) | 19.8 | 190 (90–488) |
| Entacapone +levodopa | 1067 (17.5) | 174 (76–450) | 16.7 | 491 (233–829) | 4.1 | 252 (163–446) | 8.7 | 137 (69–390) | 61.7 | 120 (60–297) |
| Ropinirole | 877 (14.4) | 294 (120–608) | 21.3 | 474 (268–939) | 45.0 | 272 (113–574) | 11.6 | 149 (71–337) | 15.3 | 172 (89–413) |
| Pergolide | 413 (6.8) | 200 (77–471) | 16.9 | 495 (250–834) | 12.8 | 108 (40–248) | 46.2 | 133 (64–311) | 14.5 | 156 (79–450) |
| Procyclidine | 268 (4.4) | 283 (122–582) | 8.2 | 219 (122–513) | 21.6 | 174 (67–526) | 46.3 | 370 (145–691) | 19.4 | 188 (132–405) |
| Bromocriptine | 194 (3.2) | 60 (42–141) | 4.1 | 217 (99–276) | 45.4 | 85 (31–263) | 43.8 | 55 (43–83) | 5.2 | 87 (44–127) |

Q1, 25[th] percentile; Q3, 75[th] percentile, N, total number of patients; PD, Parkinson's disease.

[a]The table includes medications that were prescribed in ≥1% of the patients with PD.

[b]Median time is reported for the overall population, including patients progressing to second-line treatment and those who continued on first-line treatment through the end of the study period.

[c]Continued indicates no treatment change until end of follow-up for the patient.

[d]Augmented indicates the addition of at least one PD medication to the current PD regimen.

[e]Discontinued indicates no prescription for the PD medication after the last prescription of >60 days.

[f]Remaining percentage included undefined and indeterminate cases, representing patients who received treatment<60 days or started treatment within 60 days of end of follow-up, respectively.

monotherapy, 3474 (57.9%) remained on the same treatment to the end of follow-up, 734 (12.2%) changed to polytherapy, and 275 (4.6%) switched to another monotherapy. For patients receiving first-line polytherapy (n = 92), 46 (50.0%) remained on the same treatment to the end of follow-up, 21 (22.8%) changed to a different polytherapy regimen, 16 (17.4%) switched to monotherapy, and 8 (8.7%) were undefined because they had treatment for <60 days.

For the UK study population (N = 7755), 21.4%, 15.7%, 19.4%, 11.3%, 9.2% and 23.1% of patients received none, one, two, three, four, and at least five treatment lines, respectively, over a median follow-up of 1006 (90–3797) days. Of the 4176 patients receiving first-line monother-apy, 995 (23.8%) remained on the same treatment to the end of follow-up, 1192 (28.5%) changed to polytherapy, and 717 (17.2%) switched to another monotherapy. For patients receiving first-line polytherapy (n = 990), 193 (19.5%) remained on the same treatment to the end of follow-up, 88 (8.9%) changed to a different polytherapy regimen, and 52 (5.3%) switched to another monotherapy.

For patients who had more than one treatment line, trends were examined across the next three lines of treatment. Results showed that, as with first-line treatment, monotherapy was more common than polytherapy for the second and third treatment lines (Fig 1A and 1B).

## Discussion

This retrospective analysis of data using the IBM MarketScan claims and CPRD databases involving patients with newly diagnosed PD provides insight into characteristics of the patient population and the current state of PD treatment. The study found that even though there was

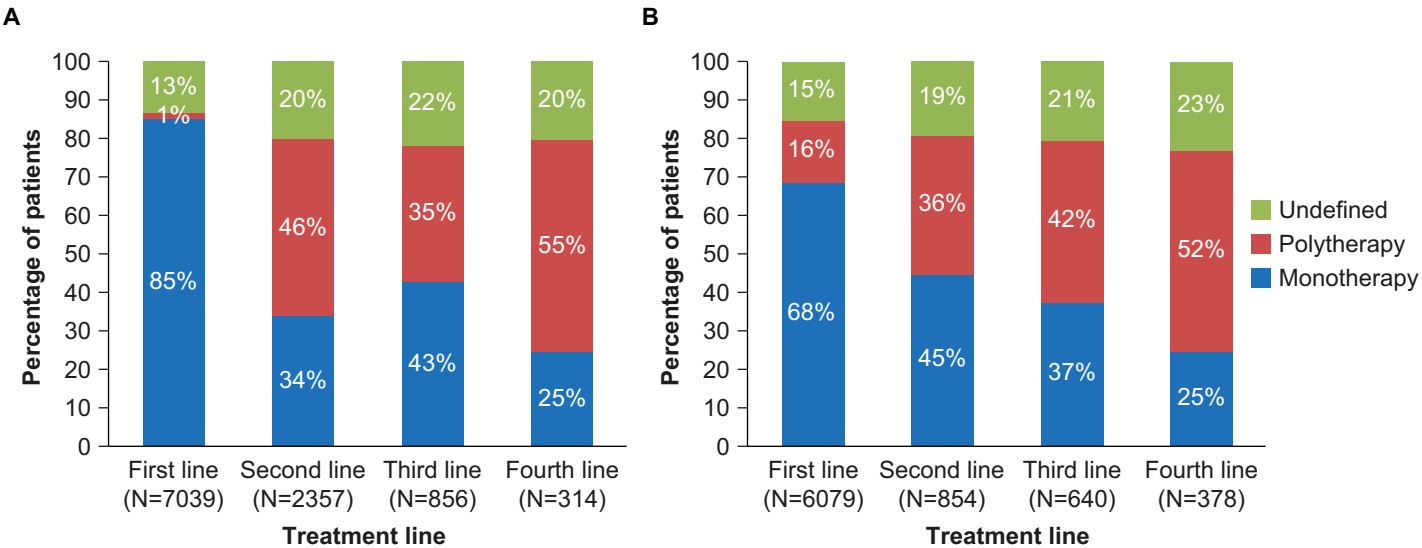

**Fig 1.** PD medication treatment lines for (A) patients with PD in the US who received one or more treatment lines, and (B) patients with PD in the UK who received one or more treatment lines.

variation in the type of comorbidities identified in the study populations, hypertension, type 2 diabetes mellitus, and ischemic heart disease were among the highly prevalent comorbidities. It is important to note these comorbidities may affect treatment decisions in PD patients owing to the potential for drug–drug interactions.

Our study also found that 37.6% of the patients in the US and 21.4% of the patients in the UK did not receive any prescription of PD medication during follow-up. A previous study conducted in the US that evaluated patients with PD from 2007 to 2010 using the Medicare database found that approximately 18% of the patients did not have PD medication prescriptions, which is lower than what we have found in this study [7]. However, they included incident and prevalent cases of PD patients, and therefore might have a higher proportion of patients with severe disease symptoms requiring treatment. The IBM MarketScan database includes information on filled prescriptions, while in the CPRD database physician prescriptions are reported. This could be why the proportion of untreated patients in higher in the IBM MarketScan database. We were not able to evaluate the reasons for not receiving treatment because this information is not included in the database, but it could be due to several factors. Although we used validated definitions to define PD cases in the IBM MarketScan database, some false positive cases may have been included in our analysis. Because we did not identify any validated algorithms for the CPRD database analysis, we used a stringent case definition for PD to minimize the number of false positive cases. However, we may have also included some false positive cases in the CPRD study population. Additionally, the study population may have included patients with less severe symptoms and therefore treatment was deemed unnecessary, because we included only newly diagnosed patients. On the other hand, these results may also reflect lack of access to care for some patients. Further research is needed to confirm our findings and assess the reason for lack of treatment in these patients.

Initial treatment was most often with monotherapy (85.2% and 68.5% of the US and UK patients, respectively). There are very limited published data available on treatment patterns in incident PD patients. Previous studies that have reported treatment patterns have included incident and prevalent PD cases [6–10]. Despite this difference, these previous studies also found that the majority of the patients were treated with monotherapy, ranging from 60% to

69.5% [6,9]. The proportion of patients who received polytherapy increased with subsequent treatment lines, which may indicate progression of disease. In the US, among patients who received a second-line treatment, more patients were prescribed polytherapy than monotherapy. In the UK, however, more patients were prescribed monotherapy as second-line treatment than polytherapy. This may be due to differences in treatment guidelines between the two countries.

Levodopa was the most frequently prescribed PD medication in our study. Previous studies have also reported levodopa as the most frequently prescribed PD medication, ranging from 37.4% to 90.0% [6,7,9,10]. It should be noted that the proportion of patients who were treated with levodopa was much higher in the US than in the UK (70.1% and 29.0% respectively). There were differences in the types of PD medications that were prescribed frequently in the US and UK.

There were 57.9% of patients in the US and 23.8% of patients in the UK who remained on first treatment until the end of the study. Changes in initial treatment, particularly addition of another medication, may reflect poor disease control. We found that 16.8% and 40.2% of patients in the US, and 45.7% and 14.0% of the patients in the UK, who were initiated on monotherapy or polytherapy, respectively, underwent treatment switches/additions from first- to second-line therapy. Although we were not able to measure disease progression, it is tempting to speculate that most patients who persisted on initial treatment had successful treatment outcomes.

Among PD medications prescribed to ≥1% of the US study population, approximately 12.3% to 40.5% of the patients augmented, 2.9% to 20.7% switched their treatment, and 8.8% to 23.5% discontinued treatment. Among PD medications taken by ≥1% of the UK study population, approximately 4.1% to 45.4% of the patients augmented, 8.7% to 46.3% switched their treatment, and 5.2% to 61.7% discontinued treatment. For each of the medications prescribed to ≥1% of the PD patients, a higher proportion of patients in the US than in the UK remained on the first-line treatment. The differences in the changes in treatment for the two study populations may be due to the duration of follow-up, which was much longer in the CRPD than in the IBM MarketScan database. Changes in treatment may be due to lack of effectiveness or adverse events, and are impacted by physician practices, cost, and availability of other treatment options. Further studies are needed to evaluate factors that contribute to the differences in prescription patterns across regions.

There were several study limitations. The data are subject to miscoding, errors in reporting, and missing information. Although our study is not entirely representative of all insurance types, it does represent a significant cross-section of insured lives in the US and, to our knowledge, is the first to report prescribing data in incident PD patients. Although the CPRD is a large database with data from primary care practitioners and fairly representative of the UK population, it might still exclude other patient populations in the study. In both databases, there is a possibility of misclassification of the PD status of some of the patients. Secondly, there might be missing information or miscoding of disorders in the databases. Thirdly, the observation period and the duration of follow-up in the two databases were different due to availability of data, which may have contributed to some of the noted differences in the treatment patterns between the two regions, as treatment practices change over time. Lastly, the data on medication are obtained from prescriptions by physicians in the CPRD database, and in the IBM MarketScan database, it is obtained for pharmacy fill records; however, there are no data to confirm that the patient took the medication. Despite these limitations, the large sample size of the databases allowed for evaluation of treatment patterns in a large sample size of patients with PD as compared with other study designs (clinical trials or prospective cohort studies). Additionally, the databases provide an opportunity to assess treatment of patients with PD in clinical practice.

## Conclusion

This study provides an important description of current real-world treatment patterns in newly diagnosed patients with PD in the US and the UK. The majority of treated patients were prescribed first-line treatment with monotherapy. As demonstrated in this study, use of large healthcare databases can provide insight on current treatment patterns to inform policy decisions and direct future at improving care for PD patients.

## Acknowledgments

The authors thank the patients whose data contributed to this research. The authors acknowledge the contribution of Knut Mueller (previous employee of UCB Pharma) for statistical programming, and Helen Ysak PhD (UCB Pharma, Atlanta, GA, USA) and Nicole Meinel PhD (Evidence Scientific Solutions, London, UK) for coordination of publication and editorial assistance, funded by UCB Pharma.

## Author Contributions

**Conceptualization:** Linda Kalilani, David Friesen, Nada Boudiaf, Mahnaz Asgharnejad.

**Formal analysis:** David Friesen, Nada Boudiaf.

**Methodology:** Linda Kalilani, David Friesen, Nada Boudiaf, Mahnaz Asgharnejad.

**Writing – original draft:** Linda Kalilani.

**Writing – review & editing:** David Friesen, Nada Boudiaf, Mahnaz Asgharnejad.

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
