## [Decision Letter · Decision Letter 0]

7 Oct 2019

PONE-D-19-24895

The characteristics and treatment patterns of patients with Parkinson’s disease in the United States and United Kingdom: A retrospective cohort study

PLOS ONE

Dear Dr Kalilani,

Thank you for submitting your manuscript to PLOS ONE. After careful consideration, we feel that it has merit but if the information required in the reviewer's comments is added to the publication, the article once published will attract much more interest from the readership of PLOS ONE and general public. Therefore, we invite you to submit a revised version of the manuscript that addresses the points raised during the review process.

We would appreciate receiving your revised manuscript by Nov 21 2019 11:59PM. To enhance the reproducibility of your results, we recommend that if applicable you deposit your laboratory protocols in protocols.io, where a protocol can be assigned its own identifier (DOI) such that it can be cited independently in the future. For instructions see: http://journals.plos.org/plosone/s/submission-guidelines#loc-laboratory-protocols

We look forward to receiving your revised manuscript.

Kind regards,

Erich Talamoni Fonoff, M.D, Ph.D. Associate Professor

Academic Editor

PLOS ONE

Journal Requirements:

Additional Editor Comments (if provided):

Interesting and quite useful data to be published about medical treatment of PD in two countries with different health care systems.

I would be quite interesting if the author could provide any type of information on two additional treatment that are sure available in both countries, any kind of surgical and Rehabilitation treatments. The time frame is also very interesting, by means of when such treatment became available in each of the countries.

Reviewers' comments:

Reviewer's Responses to Questions

**Comments to the Author**

1. Is the manuscript technically sound, and do the data support the conclusions?

Reviewer #1: Yes

Reviewer #2: Yes

2. Has the statistical analysis been performed appropriately and rigorously? 

Reviewer #1: Yes

Reviewer #2: I Don't Know

3. Have the authors made all data underlying the findings in their manuscript fully available?

Reviewer #1: Yes

Reviewer #2: Yes

4. Is the manuscript presented in an intelligible fashion and written in standard English?

Reviewer #1: Yes

Reviewer #2: Yes

5. Review Comments to the Author

Reviewer #1: I consider the article relevant. Despite small differences between the data analyzed in both countries, the results were relevant. This article concludes that it is important to build a robust database for population analysis.

Reviewer #2: The manuscript presents good methodology arguments and presents consistent results.

There is Consistency and Cohesion in the writing. It is emphasized that new research needs to be done so that it can have increasingly consistent results, especially with other methodologies.

6. PLOS authors have the option to publish the peer review history of their article (what does this mean?). If published, this will include your full peer review and any attached files.

Reviewer #1: No

Reviewer #2: No

---

## [Author Response · Author response to Decision Letter 0]

7 Nov 2019

Please see the 'Response to Reviewers' document for our responses to each point raised by the Editor and reviewers.

---

## [Editor Report · Decision Letter 1]

12 Nov 2019

The characteristics and treatment patterns of patients with Parkinson’s disease in the United States and United Kingdom: A retrospective cohort study

PONE-D-19-24895R1

Dear Dr. Kalilani,

We are pleased to inform you that your manuscript has been judged scientifically suitable for publication and will be formally accepted for publication once it complies with all outstanding technical requirements.

With kind regards,

Erich Talamoni Fonoff, M.D, Ph.D. Associate Professor

Academic Editor

PLOS ONE

Additional Editor Comments (optional):

Accept
---

## [Editor Report · Acceptance letter]

15 Nov 2019

PONE-D-19-24895R1 

The characteristics and treatment patterns of patients with Parkinson’s disease in the United States and United Kingdom: A retrospective cohort study 

Dear Dr. Kalilani:

I am pleased to inform you that your manuscript has been deemed suitable for publication in PLOS ONE. Congratulations! Your manuscript is now with our production department. 

With kind regards,

on behalf of

Prof. Erich Talamoni Fonoff 

Academic Editor

PLOS ONE